# Transcriptomic analysis of potato (*Solanum tuberosum* L.) tuber development reveals new insights into starch biosynthesis

**Maryam Shirani-Bidabadi[1], Farhad Nazarian-Firouzabadi** [1]*, **Karim Sorkheh[2], Ahmad Ismaili[1]**

**1** Production Engineering and Plant Genetics Department, Faculty of Agriculture, Lorestan University, Khorramabad, Iran, **2** Production Engineering and Plant Genetics Department, Faculty of Agriculture, Shahid Chamran University of Ahvaz, Ahvaz, Iran

* nazarian.f@lu.ac.ir

**Data Availability Statement:** The datasets used and/or analyzed during the current study are presented in the paper and its Supporting Information files, available as S1-S7 Tables. The

## Abstract

Potato tubers are rich sources of various nutrients and unique sources of starch. Many genes play major roles in different pathways, including carbohydrate metabolism during the potato tuber's life cycle. Despite substantial scientific evidence about the physiological and morphological development of potato tubers, the molecular genetic aspects of mechanisms underlying tuber formation have not yet been fully understood. In this study, for the first time, RNA-seq analysis was performed to shed light on the expression of genes involved in starch biosynthesis during potato tuber development. To this end, samples were collected at the hook-like stolon (Stage I), swollen tips stolon (Stage II), and tuber initiation (Stage III) stages of tuber formation. Overall, 23 GB of raw data were generated and assembled. There were more than 20000 differentially expressed genes (DEGs); the expression of 73 genes involved in starch metabolism was further studied. Moreover, qRT-PCR analysis revealed that the expression profile of the starch biosynthesis DEGs was consistent with that of the RNA-seq data, which further supported the role of the DEGs in starch biosynthesis. This study provides substantial resources on potato tuber development and several starch synthesis isoforms associated with starch biosynthesis.

## Introduction

Global potato (*Solanum tuberosum* L.) production has increased over the past decades [1], and considering the growing need for food in the future, the necessity for the increased production of potatoes continues to rise [2]. In addition to food and feed, potato tubers are also the starting material for the next generation when used as so-called seed tubers. Therefore, processes associated with tuber formation, storage, sprouting, and development have been intensively studied over the recent decades [3–7]. Among many aspects of studying tuber development, there has been a great deal of effort focusing on unraveling the nature of stimuli and molecular mechanisms underpinning tuber growth and development [8].

sequencing database could also be downloaded from NCBI under the accession number PRJNA530118.

**Funding:** The authors received no specific funding for this work.

**Competing interests:** The authors have declared that no competing interests exist

Tuberization in potatoes typically begins from lateral underground buds located at the base of the main stem, forming specialized underground stems called stolons [9]. At the onset of tuberization, longitudinal stolon growth stops and the sub-apical region of the stolon begins to swell [10]. To form mature tubers, restricted longitudinal stolon growth followed by cell division and growth at random orientations continues until the tuber reaches its full size. Despite accumulated data regarding various environmental and endogenous factors influencing tuberization, the molecular genetic aspects of mechanisms controlling tuber formation are still largely unclear. To this end, extensive molecular genetic studies have been carried out on developing potato tubers. It is obvious that the developmental changes and processes occurring during different stages of potato tuber development are influenced by several endogenous and exogenous factors including sugar molecules, plant hormones, and environmental cues [11]. For instance, the expression of genes coding for main storage proteins such as patatin and genes encoding starch biosynthesis are differentially regulated during the tuber life cycle [12–14]. Furthermore, Ewing et al. [15] revealed the Quantitative trait loci (QTLs) associated with tuberization and tuber dormancy, suggesting a firm molecular genetic basis for the identification of key phytohormones involved in morphological and biochemical processes.

The starch biosynthesis pathway is a biochemical reaction occurring during tuber development [16]. Starch is synthesized inside potato amyloplasts as a water-insoluble mixed polymer, and it is composed of amylose ($\sim 25\%$), an essentially linear polymer of $\alpha$-1,4-linked glucose moieties, and a branched amylopectin polymer of $\alpha$-1,4-linked glucose units with $\alpha$-1,6-linked glucose branches [17, 18]. The pathway of starch biosynthesis occurring in potato tubers involves the synthesis of ADP-glucose (ADP-Glc) by AGPase from glucose-1-phospate (Glc1P) followed by the coordinated reactions of $\alpha$-1,4-linked glucan chain elongation by starch synthase (SS), branching at $\alpha$-1,6 positions, and the debranching of specific branch linkages by starch branching enzymes (SBEs) and debranching enzymes (DBEs), respectively. Each class of starch biosynthesis enzymes is divided into different subunits and isoforms.

In potatoes, genomic, cDNA, and expressed sequence tag (EST) sequences encoding starch synthase genes can be found in databases, and the sequence data of the whole genome of some potato species are publicly available. Therefore, conducting a comprehensive expression analysis of multiple gene families in potato to shed light on the starch biosynthesis pathway is favorable. Many candidate genes associated with starch biosynthesis and metabolism have been widely studied in potatoes. However, the key genes, the exact number of genes, and how their expression levels are regulated during starch biosynthesis in developing potato tubers vary in different studies. For instance, microarray analysis in potatoes shows that many DEGs directly involved in the sucrose-to-starch synthesis pathway were strongly up-regulated during tuber development [14]. Although microarray analysis was able to prove the function of many starch synthase genes, the accuracy of expression measurements for low-abundance transcripts with variable hybridization properties of probes is a major limitation of this technique [19]. Furthermore, arrays are limited to interrogating transcripts/genes by using relevant probes on the array. ESTs and microarray-based analyses have been carried out, deciphering genes involved in regulating main biosynthetic pathways, including tuber development and starch biosynthesis and degradation [14, 20]. However, neither EST nor microarray analysis has fully utilized the available sequence data information. While such methods have some major limitations [21], given the availability of potato genomic sequence data, robust whole-genome analysis techniques such as RNA-sequencing (RNA-seq) have not been used to study tuberization.

High-throughput sequencing, especially RNA-seq technology, has overcome many limitations of past molecular approaches. RNA-seq analysis enables researchers to simultaneously analyze tens of thousands of transcripts for gene discovery and transcript frequencies at different time points and under certain conditions. Besides, despite the

provision of comprehensive information on the transcriptome, RNA-seq requires no prior knowledge of genome content [22, 23].

Although several studies have dealt with tuber developmental stages, to the best of our knowledge, there is no complete transcriptome analysis on potato tubers during developmental stages by RNA-seq technology. In this study, the differential expression of genes during main potato tuber development stages is analyzed and discussed. An immediate application of transcriptome sequence data of tuber development stages includes gene expression profiling at different tuber developmental stages.

## Materials and methods

### Plant materials

Tubers of potato Marfona cultivar ($\sim$35 gram), a popular early baker, high-yielding, resistant to dry rot, and susceptible to potato cyst nematodes, were planted in 25 cm × 25 cm × 30 cm pots containing sterile sandy-loam soil at the end of September 2016 under 12h/12h light/dark conditions at $24 \pm 2°C$ in the greenhouse of Faculty of Agriculture, Lorestan University, Khorramabad, IRAN. For RNA-seq analysis of gene expression during tuber development, ten growing stolon samples from three stages: hook-like tip (Stage I), swollen tips (Stage II), and tuber initiation (Stage III) were collected six weeks after planting (Fig 1). Tuber samples were promptly frozen in liquid nitrogen and stored at -80°C until RNA extraction.

### RNA extraction, cDNA library, and Illumina sequencing

Total RNA was extracted from tubers, according to Chang et al. [24]. The RNA quality and quantity were assessed and measured by 1.0% agarose gel electrophoresis and NanoDrop (Beijing, China). RNA integrity analysis and quantitation were also carried out by the Agilent 2100 Bioanalyzer system (Agilent Technologies Co. Ltd., Beijing, China). Illumina library preparation and sequencing were done by Novogene Bioinformatics Technology Company (Beijing, China). Out of 18 samples checked for RNA qualitative and quantitative characteristics, six samples were chosen by Novogene Bioinformatics Technology Company for RNA sequencing

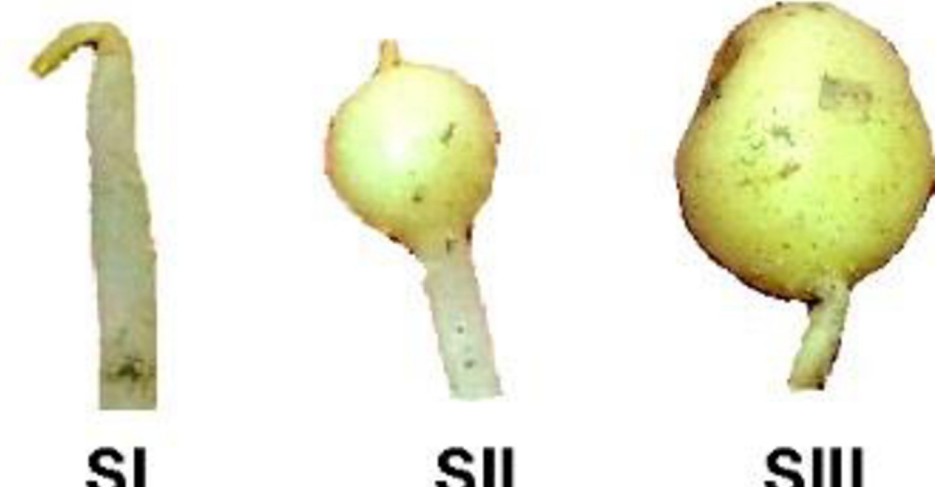

**Fig 1. Illustration of developmental stages of potato tubers used for RNA-seq analysis.** Developmental stages were chosen according to Kloosterman et al. (2005). SI) stage I: hook-like tip, SII) stage II: swollen tips, and SIII) stage III: tuber initiation.

based on the RNA integrity number (RIN>7). Sequencing was done with Illumina HiSeq 2500, using paired-end $2 \times 150$ bp reads.

## Bioinformatics analysis of RNA sequences

The quality of the primary raw sequences was checked by FastQC [25]. To accurately analyze the RNA-Seq data, Trimmomatic version 0.36 was used to remove poor quality sequences and to trim raw reads [26]. The release-34 version of the *Solanum tuberosum* reference genome (ftp://ftp.ensemblgenomes.org/pub/release-34/plants/fasta/solanum_tuberosum/dna/) was placed against sequenced reads by Bowtie 2 (v2.3.3.1) to obtain an overall alignment rate [27]. RNA-seq by expectation maximization (RSEM v1.3.0) [28] was used for quantifying the expression of each transcript and each gene [29]. The DESeq2 package (Bioconductor) calculated differential expression based on the negative binomial distribution model [30]. To identify differentially expressed genes (DEGs), log2 fold change more than or equal to 1 (as up-regulated genes) and less than or equal to -1 (as down-regulated genes) with FDR $\leq 0.05$ was considered. Blast2GO was used to BLAST the DEGs for functional annotations against GO and NCBI nr databases [31]. Gene ontology (GO) of DEGs was performed using the AgriGO v2.0 [32]. Further analysis was accomplished by KEGG (https://www.kegg.jp/kegg/tool/annotate_sequence.html).

## Real-time PCR analysis

Total RNA was isolated by the lithium chloride method [24], followed by DNase I, RNase-Free (SinaClon BioScience, Cat. No:MO5401) treatment, and purified RNA was dissolved in DEPC-treated water. The quality and quantity of purified RNA were determined as described. cDNAs were synthesized using Thermo Scientific RevertAid First Strand cDNA Synthesis Kit (The thermo Fisher Scientific Inc., #k1622). To design RT primers (S1 Table in S1 Data), sequences were obtained from Ensemble in the FASTA format and deposited in Primer3 (http://primer3.ut.ee/). Parameters were set according to Thornton and Basu [33]. This was done by Beacon designer$^{TM}$ as the second step. In order to validate the RNA-Seq reads, 6 DEGs associated with starch synthesis and tuber development were picked based on KEGG for qRT-PCR analysis. Overall, six DEGs including granule-bound starch synthase 1, soluble starch synthase I, soluble starch synthase II, ADP glucose pyrophosphorylase large subunit 2, endo-1,4-β-glucanase, and cellulose synthase D were selected for qRT-PCR analysis. Among different internal control genes tested, the *ef*1α gene (PGSC0003DMG400023270) [2, 34, 35] was found stable at different developmental stages and was hence chosen as the internal control for qRT-PCR analysis. The qRT-PCR was done by SYBR® Premix Ex Taq$^{TM}$ II (Tli RNase H Plus, Cat#RR820Q) with two technical and two biological replications for each sample. Real-time PCR was carried out by an Applied Biosystems apparatus (ABI, Biosystem, USA).

PCR cycles were as follows: initial denaturation at 95˚C for 3 min, followed by 40 cycles of denaturation at 95˚C (10 to 15 sec), annealing at 55–60˚C (15 to 30 sec), and extension at 72˚C (30 sec). qRT-PCR data were analyzed, using the $2^{-\Delta\Delta ct}$ method [36].

## Results

### Alignment of Illumina sequencing data to the potato reference genome

Since the intial release of the potato genome sequence in 2011 [37], much effort has been made to decipher central pathways in potato cells including those of carbohydrate methabolisem. To identify genes involved in starch biosynthesis during tuber development, cDNA libraries of three developmental stages (Fig 1) were sequenced. RNA reads were submitted at SRA under

BioProject accession number PRJNA530118. Nearly 45 million reads were produced per sample, encompassing almost 23 GB of sequence data, sufficient for the quantitative gene expression analysis. Almost 0.1% of Illumina reads, dropped at the trimming step. The sequence reads were aligned to the potato reference genome database. Of the 44716722 reads, more than 50% (51.21% to 58.86%) matched a unique potato genomic location and nearly 10% (10.43% to 18.61%) to multiple potato genomic locations (Table 1). Mapping with reference genome illustrated an overall 68.48% alignment on average. At the counting stage, 38790 genes and 55726 transcripts were extracted.

## Differentially expressed genes

In this study, the DEGs associated with starch synthesis during potato tuber development were identified. The total number of genes counted in the developing tubers was 25914, representing almost 64% of the annotated transcriptome of the potato genome. Of these 25914 genes, 23697, 23745, and 23269 genes were expressed in stage I vs stage II, stage I vs stage III, and stage II vs stage III, respectively. Fig 2 illustrates the number of genes uniquely expressed in each developmental stage as well as the number of genes that were shared with one or two other stages. Overall, 16646 genes were expressed in the three stages, with 794, 240, and 436 unique genes in the first, the second, and the third stage, respectively (Fig 2). When contrasting expression at different tuber development stages, the first two stages with 635 genes, had the highest number of co-expressed genes, followed by 234 genes for the first and the third stages and then 315 genes for the second and the third stages (Fig 2).

Fig 3 shows changes in gene expression profile through the three developmental stages. Genes were considered up-regulated (FDR $\leq$ 0.05, Log2FC $\geq$ 1) and down-regulated (FDR $\leq$ 0.05, Log2FC $\leq$ -1) as DEGs or transcripts (S2.1-S2.3 and S3.1-S3.3 Tables in S1 Data). As can be seen, the number of up-regulated genes and down-regulated genes varied considerably in different stages (Fig 3). The highest number of both up-regulated and down-regulated genes were found in stage I vs stage III. Since more genes were significantly increased or decreased, the changes that occurred when progressing from stage II to stage III were more pronounced in comparison with the changes that occurred when progressing from stage I to stage II.

## Classification of potato genes

Differentially expressed genes (-1 $\geq$ Log2FC $\geq$ 1, FDR $\leq$ 0.01) were chosen and deposited in KEGG (https://www.kegg.jp/kegg/tool/annotate_sequence.html). KEGG mapper reconstruction resulted in 46, 188, and 89 pathways for stage I vs II, stage I vs III, and stage II vs III, respectively. In all stages, a large number of genes corresponding to carbohydrate, energy, amino acids, lipid and protein metabolism, environmental informational processing, genetic information processing, cellular processes, signaling, and cellular process were found (Fig 4). Comparisons of up-regulated and down-regulated genes through three developmental stages

**Table 1. Summary of read numbers based on the RNA-Seq data analysis from developing potato tuber stages.**

| | Stage I/R1 | Stage I/R2 | Stage II/R1 | Stage II/R2 | Stage III/R1 | Stage III/R2 |
|---|---|---|---|---|---|---|
| **Total Reads** | 46934762 | 46430834 | 42249890 | 42509714 | 43026960 | 47148172 |
| **Mapped** | 32735482 (69.8%) | 31372258 (67.5%) | 28724276 (68%) | 28809216 (67.8%) | 30041441 (69.8%) | 32056028 (68%) |
| **Unique Match** | 27626799 (58.9%) | 24679506 (53.1%) | 23082968 (54.6%) | 24377196 (57.3%) | 22034189 (51.2%) | 25346433 (53.8%) |
| **Multi-Position Match** | 5108683 (10.9%) | 6692752 (14.4%) | 5641308 (13.3%) | 4432020 (10.4%) | 8007252 (18.6%) | 6709595 (14.2%) |
| **Unmapped** | 14199280 (30.3%) | 15058576 (32.4%) | 13525614 (32.0%) | 13700498 (32.2%) | 12985519 (30.2%) | 15092144 (32.0%) |

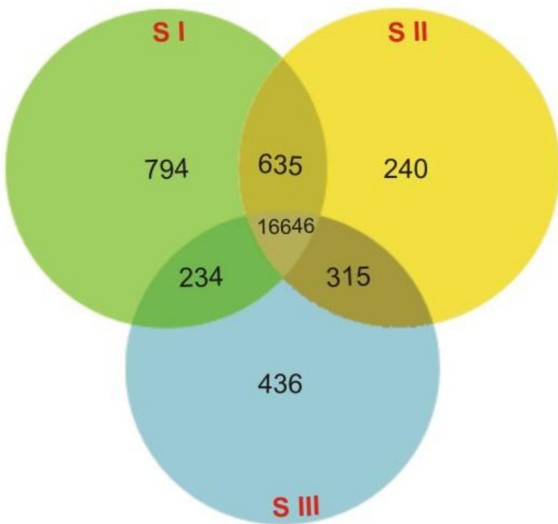

**Fig 2. Venn diagrams representing the number of differentially expressed genes (DEGs) in three developmental stages in potato tubers.**

revealed that the genetic information process, carbohydrate metabolism, and lipid metabolism genes were more abundant than other expressed genes. Carbohydrate-metabolism-associated genes were found in all three comparisons (Fig 4). Detailed information regarding comparing the gene expression profile is presented in the S4.1-S4.3 Table in S1 Data.

DEGs profile are presented in S5 Table in S1 Data. All GO terms are presented in the S6 Table in S1 Data. GO terms were categorized in biological process, molecular function, and cellular component categorizes with many subclasses (Table 2).

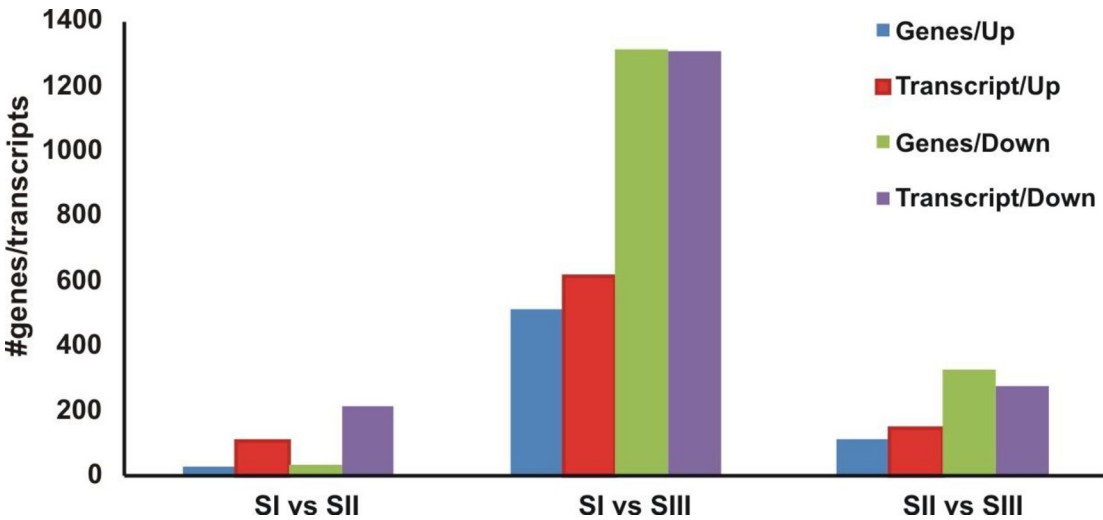

**Fig 3. Number of differentially expressed genes and transcripts in the three developmental stages.** The number of up-regulated and down-regulated genes between different stages are summarized.

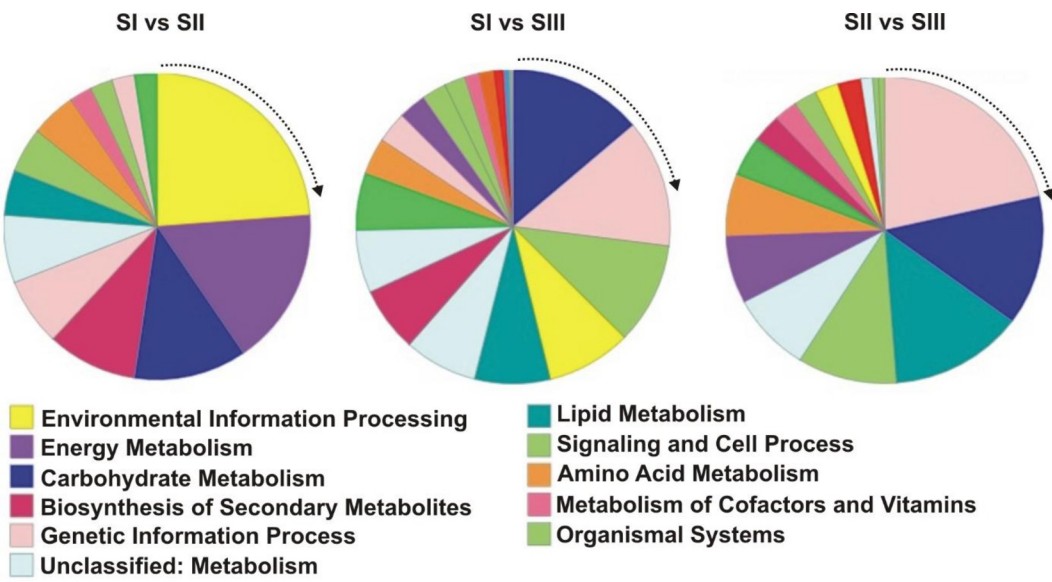

**Fig 4. Functional annotation of the RNA-seq assembly-unique sequences for genes whose putative function could be determined based on BlastKOALA in the KEGG database.**

### Differential gene expression profile associated with starch synthesis

The DEGs of the three tuber developmental stages were analyzed to identify the genes involved in starch synthesis. The RNA-seq analysis resulted in the identification of 73 genes/transcripts associated with starch biosynthesis pathways (Fig 5). Among the 73 starch-associated genes, the number of up-regulated genes decreased as tuber development progressed, whereas the number of down-regulated genes increased from 20 in stage I to 33 in stage III (S7 Table in S1 Data). DEGs encoding 73 enzymes have roles in starch biosynthesis in developing tubers in the cytosol and the plastid. The enzymes include sucrose synthase, starch branching enzyme, starch synthase (I-VI), granule-bound starch synthase I, cell wall invertase, etc.

Chromosomal positions of genes involved in starch biosynthesis were retrieved from the Spud DB genome browser v4.03 and depicted [38] (Fig 5). In silico, the mapping of genes associated with starch biosynthesis on potato chromosomes showed an uneven distribution of the genes on all 12 chromosomes. Chromosome 7 with 11 genes and chromosome 11 with three genes had the highest and the lowest numbers of starch-associated genes, respectively (Fig 5 and S7 Table in S1 Data). Furthermore, the patterns of gene distribution on individual potato

**Table 2. Segmentation of GO terms for three main groups and their sub-groups.**

| Cellular component (GO:0005575) | Biological process (GO:0008150) | Molecular function (GO:0003674) |
|---|---|---|
| 1. Extracellular region (GO:0005576) | 1. Response to stimulus (GO:0050896) | 1. Electron carrier activity (GO:0009055) |
| 2. Membrane part (GO:0044425) | 2. Developmental process (GO:0032502) | 2. Structural molecule activity (GO:0005198) |
| 3. Membrane (GO:0016020) | 3. Multicellular organismal process (GO:0032501) | 3. Nutrient reservoir activity (GO:0045735) |
| 4. Macromolecular complex (GO:0032991) | 4. Cellular component organization of biogenesis (GO:0071840) | 4. Binding (GO:0005488) |
| 5. Cell (GO:0005623) | 5. Single-organism process (GO:0044699) | 5. Nucleic acid binding transcription factor activity (GO:0001071) |
| 6. Cell part (GO:0044464) | 6. Cellular process (GO:0009987) | 6. Catalytic activity (GO:0003824) |
| 7. Organelle (GO:0043226) | 7. Metabolic process (GO:0008152) | 7. Antioxidant activity (GO:0016209) |
| 8. Organelle part (GO:0044422) | 8. Biological regulation (GO:0065007) | 8. Transporter activity (GO:0005215) |
| 9. Supramolecular fiber (GO:0099512) | 9. Regulation of biological process (GO:0050789) | 9. Molecular function regulator (GO:0098772) |
| | 10. Positive regulation of biological process (GO:0048518) | |
| | 11. Localization (GO:0051179) | |

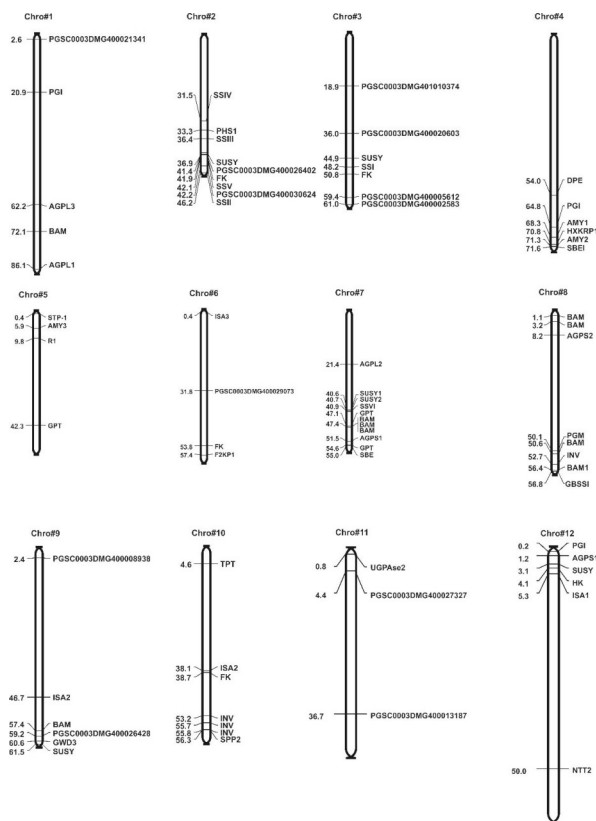

**Fig 5. Starch-synthesis-associated genes distributed on potato genome chromosomes.** The number on the left refers to centiMorgans (cM).

chromosomes also revealed that certain physical regions have a relatively higher accumulation of gene clusters. While the majority of the 73 starch associated genes were distributed over all 12 chromosomes of potato; chromosome two with four SS-encoding genes seemed to play a major role in starch synthesis in potato tubers.

## Experimental validation of RNA-Seq data

Contrary to standard microarray analysis of gene expression, the RNA-Seq technique enables the differential expression profile of expressed genes via transcript abundance with a high sensitivity for lower-abundance expressed transcripts [39]. To confirm and validate the RNA-Seq results, a number of differentially expressed genes were subjected to qRT-PCR analysis (S1 Table in S1 Data). Major starch synthase genes were up-regulated during tuber development. The expression of GBSSI, SSI, and SSII increased in SII and SIII relative to SI. However, it remained steady during starch development (Fig 6). The expression of ADP glucose pyrophosphorylase increased in SII but decreased in SIII. Endo-2,4-beta glucanase and cellulose synthase expression were elevated in SI and SII; however, the mRNA transcript level of Endo-2,4-beta glucanase was significantly increased in SIII. The correlation coefficient with elf1α as the internal control, was 0.78 (P -Value = 0.008), suggesting the significant correlation of the RNA-Seq data with that of the qRT-PCR data.

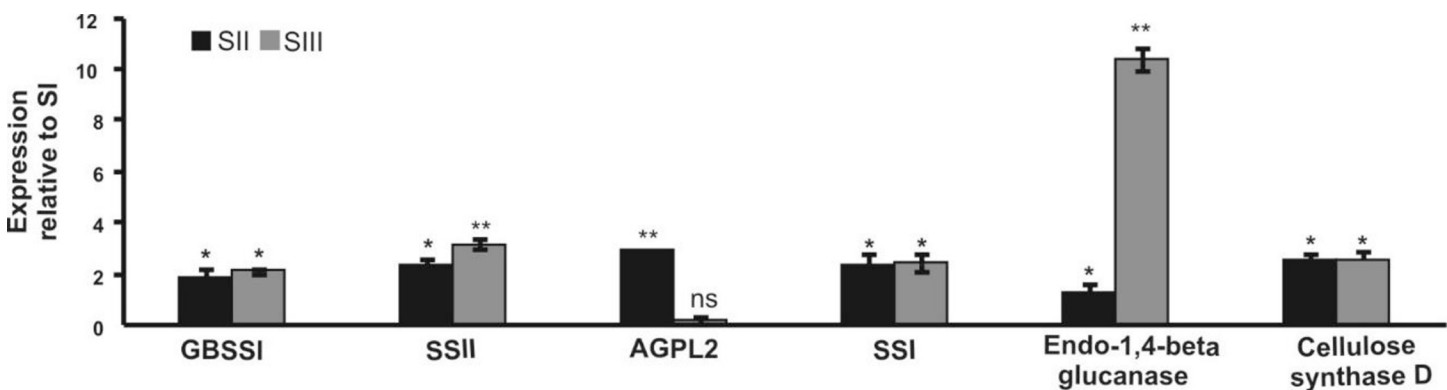

**Fig 6. qRT-PCR analysis of some Potato genes at development stages II and III relative to stage I (SI).** Espression levels were compared to SI using student t-test. *, ** represent significant at P<0.05, P<0.01 probability level, respectively. ns: indicates non-significant difference relative to SI.

## Discussion

Transcriptome data, especially RNA-seq analysis, have provided researchers with a comprehensive overview of the participation of many gene families in plant developmental stages. The outcome of such high-throughput techniques has resulted in the identification of new genes involved in various cell pathways and the confirmation of previously known gene families.

The carbohydrate group that contains genes dedicated mainly to the synthesis of starch showed an up-regulating trend when progressing from SI to SIII (Fig 7). The up-regulated profile of such starch synthesis genes was noticed at the initial stages. Later, a relatively high expression level was maintained until the beginning of tuber formation. This can be explained by the fact that starch synthesis increases at tuber induction (SI) and subsequent tuber growth and development (SII to SIII). This is evident by the fact that many genes directly involved in the starch synthesis pathway were significantly up-regulated from the swelling stolon stage onwards (Fig 7). The up-regulation of such genes has also been noticed in microarray analysis of tuber development in potatoes [14]. This can be explained by the fact that during tuber initiation (SI) and subsequent tuber organogenesis (SII and SIII), higher levels of sucrose import coincided with increased starch biosynthesis [14].

In this study, developmental-regulated genes associated with starch biosynthesis were identified among mRNA transcripts of DEGs. The process of starch biosynthesis begins with sucrose transportation from the apoplast to the cytosol. Sucrose synthase (SuSy, EC 2.4.1.13) converts sucrose to UDP-glucose and fructose, and subsequently, UDP-glucose is converted to ADP-glucose by ADP-glucose pyrophosphorylase [17]. SuSy enzymes are key players in the potato tuber starch content [40]. Similar to *Arabidopsis* [41], our RNA-seq analysis uncovered six SuSy genes that varied in transcripts 1 to 5 (S7 Table in S1 Data). In contrast to our results, genome-wide *in silico* analysis found five SuSy isoforms in the potato genome [42]. The expression of different Susy genes was up-regulated 0.2 to 5.2-fold in different developmental stages. Among six SuSy genes, one SuSy gene (PGSC0003DMG400002895) expression was at least 3-fold higher than other SuSy genes (S7 Table in S1 Data). Similarly, Van Harsselaar et al. (2017) showed that the expression of SuSy4 (with exact PGSC Gene ID: PGSC0003DMG400002895) in potato tubers was almost 8-fold higher in tubers than leaves. BLAST analysis also revealed that this SuSy gene is 90.3% similar to that of SUS4 in *Arabidopsis*. Interestingly, the expression of SUS4 was much higher in *Arabidopsis* organs as compared to the other five SuSy genes [41].

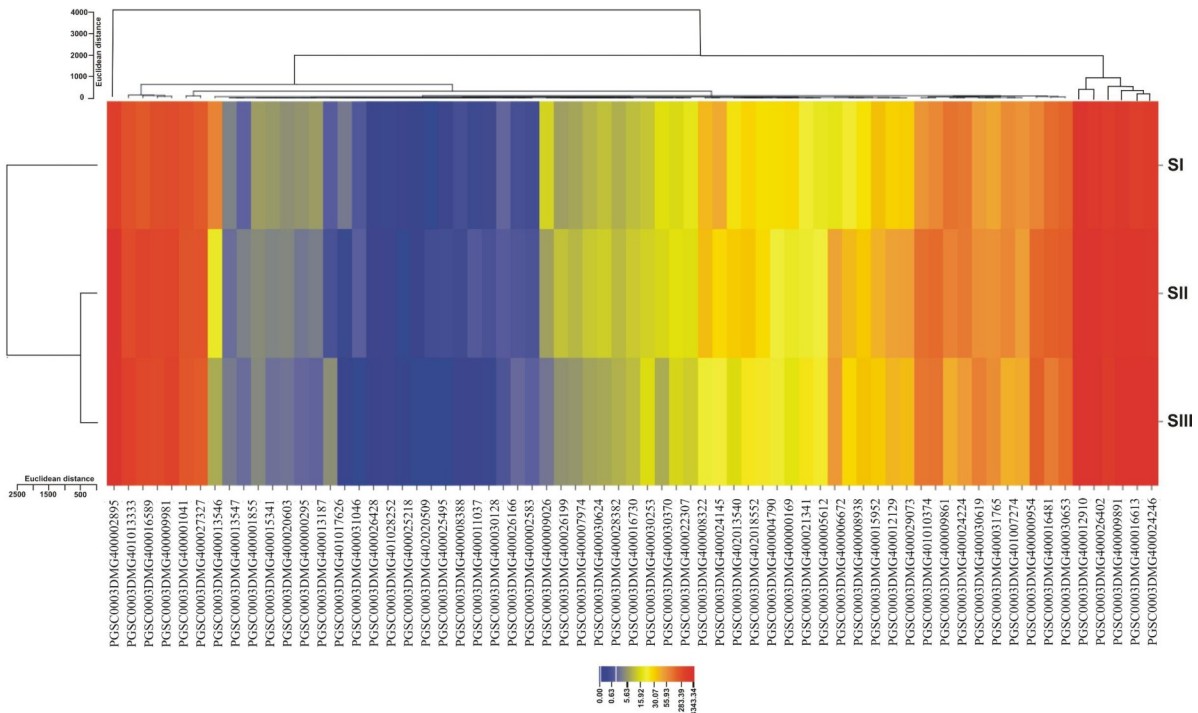

**Fig 7. Hierarchical clustering for starch metabolism involved in differentially expressed genes between stages.** The colors blue and red represent low and high expression, respectively. Columns represent different tuber development stages, and rows represent starch-associated genes. Two-dimensional-hierarchical clustering classifies 73 DEGs into three expression cluster groups according to the similarity in their expression profiles.

In plants, five isoforms of starch synthases have been reported so far: granule-bound starch synthase 1 (GBSSI) and soluble starch synthases (SSI, SSII, SSIII, and SSIV) [43]. Genome-wide in silico analysis of the potato genome showed that the potato genome contains seven starch synthases (GBSSI, GBSSII, and SSI to SSV) isoforms [42]. This is particularly interesting, because experimental examinations had not previously found SSV and SSVI isoforms. In this study, the expression of two isoforms of starch synthase, SSV and SSVI was particularly interesting, because evidence of their participation to starch biosynthesis is sporadic.

Although SSs play unique roles in starch biosynthesis, some of them may have overlapping functions [44]. The expression of all identified SS genes was up-regulated, but SSV expression was significantly higher than other SSs (S7 Table in S1 Data). The SSV gene is conserved in plants, and it is expressed during grain filling in maize [45]. Thus, it may play a role in starch synthesis, but the exact role of this enzyme remains to be investigated [45]. Interestingly, SSV lacks the C-terminal X-X-G-G-L motif conserved within other SSs [17], suggesting that SSV protein is not catalytically active [45]. The up-regulation of SSV suggests that this protein may play a role in protein-protein, protein-glucan, or glucan-glucan interactions, as suggested for ESV1 and LESV in *Arabidopsis* [46]. The expression of SSII changed during tuber developmental stages by 1.8, 2.5, and 1.4 for SI vs SII, SI vs SIII, and SII vs SIII, respectively (S7 Table in S1 Data). SSII activity in amylopectin biosynthesis overlaps with SSIII function [17] in potato tubers. The repression of SSII in many plants studied so far has shown somewhat similar phenotypes to the amylopectin structure.

Granule-bound starch synthase (GBSS), or the Waxy gene, represents two different enzymes as GBSSI and GBSSII, with GBSSI located on chromosome 8 [47]. The results of the present study revealed that the expression of GBSSI did not significantly differ in the three

developmental stages, suggesting continuous involvement of GBSSI in amylose synthesis. Although it is anticipated that amylose is synthesized downstream of amylopectin [17], the progressive mode of activity and constant expression of GBSSI indicate that amylose biosynthesis occurs side by side with amylopectin synthesis.

The expression of disproportionating enzyme (D-enzyme) a plastidial alpha-1,4-glucano-transferase, was up-regulated during potato tuber development stages (S7 Table in S1 Data). In rice endosperms, D-enzyme participates in starch synthesis by transferring maltooligosyl moieties from both amylose and amylopectin polymers to amylopectin [48]. Similar to the expression pattern in developing seeds of rice, the expression of this enzyme in developing potato tubers began at stolon initiation (SI) and then increased rapidly at SII and dropped at SIII (Fig 7 and S7 Table in S1 Data). This expression pattern is consistent with the mRNA transcripts profile of D-enzyme in developing and mature tubers [49].

The potato genome contains two isoforms of SBE confirmed by genome-wide *in silico* analysis [42]. Experimentally, the activity of two isoforms (SBEI and SBEII) of starch-branching enzymes has been shown in potato amyloplast. Although SBEI is the major isoform of SBE in potatoes, antisense down-regulation of SBEI did not lead to the alteration of the amylose content [50, 51]. In contrast, a simultaneous down-regulation of both SBEII and SBEIII in the potato tubers led to an increase in amylose content [52]. SBEIII expression in developing tubers was higher than that of SBEII (S7 Table in S1 Data). Unlike Van Harsselaar et al. [42], RNA-seq data analysis did not reveal further paralogous isoforms for any of the two SBEs in potato genome during tuber development stages. However, reports show a third SBE (Sotub07g029010.1.1) whose exact role remains to be understood [42, 53].

Starch-debranching enzymes, isoamylases (*Isa*), play a distinct role in granule initiation [54, 55]. Four different isoforms of isoamylase were differentially expressed during different tuber developmental stages (S7 Table in S1 Data), suggesting the involvement of more than three isoforms previously reported [56]. In potatoes, most of the isoamylase activity in the tubers is assigned to a heterodimer comprising of 3 isoforms [54]. Since isoamylases are involved in granule initiation and antisense, the down-regulation of both isoforms of potato isoamylases led to the reduction, but not elimination, of starch granule formation [57], suggesting that the other newly identified isoforms in this study may also play a role in granule initiation. The identification of new isoamylase isoforms can be explained by the fact that the formation of the stolon and its tuberization coincide with starch granule formation, in which isoamylases play important roles [42]. Two new isoamylase isoforms, *Isa*3 and *Isa*4, are located on chromosomes 6 and 10, respectively, whereas Isa1 and Isa2 are large proteins located on chromosomes 12 and 9, respectively (S7 Table in S1 Data).

Starch phosphorylation mainly affects starch properties such as size and morphology, and it influences starch rheological properties by affecting the activities of some starch metabolic enzymes in plants; hence, phosphorylation is an essential process demanded by the starch industry [58, 59]. Glucose moieties are normally phosphorylated at the C6 and C3 positions by glucan water dikinase (GWD) and phosphoglucan water dikinase (PWD), respectively [60]. As far as literature is concerned, there has been two GWD homologues, called R1, described in potatos [61].

In conclusion, an immediate application of our RNA-seq analysis data included gene expression profiling at the three different tuber development stages. We have gained a complete transcriptome of *Solanum tuberosum* L. during tuber developmental stages. In thousands of GBs of gene data, one can examine different metabolic pathways and their interrelationships. This study contributes to a better understanding of genes that have not yet been identified or proven to play roles in metabolic pathways. Among various cell pathways, the starch biosynthesis pathway is a central pathway involving the concerted action of many proteins and

enzymes with different cytosolic and plastidic isoforms [62]. Although isoforms have different properties and play various roles [28, 59], they also tend to participate in complex protein-protein interactions [63]. Interestingly, a significant correlation was found between qRT-PCR analysis of the starch biosynthesis DEGs with that of the RNA-seq data, which further supported the role of the DEGs in starch biosynthesis. It is worth noting that our study provides important genetic resources on potato tuber development, more specifically on starch synthesis isoforms associated with starch biosynthesis.

## Supporting information

**S1 Data. S1-S7 Tables illustrating RNA-seq data analysis during potato tuber development.**
(XLS)

## Acknowledgments

We greatly thank Dr. Ahmad Tahmasebi, and Engineer Morteza Hadizadeh whose valuable advices improved the analysis. The authors would also like to thank Dr. Mahmood Naderan-Tahan for providing the terms of use at Shahid Chamran University High-Performance Computing Center.

## Author Contributions

**Conceptualization:** Farhad Nazarian-Firouzabadi.

**Data curation:** Maryam Shirani-Bidabadi, Karim Sorkheh, Ahmad Ismaili.

**Formal analysis:** Maryam Shirani-Bidabadi, Karim Sorkheh, Ahmad Ismaili.

**Investigation:** Ahmad Ismaili.

**Supervision:** Farhad Nazarian-Firouzabadi.

**Writing – original draft:** Maryam Shirani-Bidabadi.

**Writing – review & editing:** Farhad Nazarian-Firouzabadi.

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
