## [Decision Letter · Decision Letter 0]

30 Oct 2023

PONE-D-23-23589Transcriptomic analysis of potato (Solanum tuberosum l.) tuber development reveals new insights into starch biosynthesisPLOS ONE Dear Dr. Nazarian‐Firouzabadi,

Thank you for submitting your manuscript to PLOS ONE. After careful consideration, we feel that it has merit but does not fully meet PLOS ONE’s publication criteria as it currently stands. Therefore, we invite you to submit a revised version of the manuscript that addresses the points raised during the review process.

ACADEMIC EDITOR: I request authors to follow the reviewer's suggestions and revise the manuscript. 

We look forward to receiving your revised manuscript.

Kind regards,

Karthikeyan Adhimoolam

Academic Editor

PLOS ONE

“This research was partly supported by a grant from Lorestan University.”

“The authors received no specific funding for this work”

5. We are unable to open your Supporting Information file [Supporting Information_Plos one.rar]. Please kindly revise as necessary and re-upload.

Reviewers' comments:

Reviewer's Responses to Questions

**Comments to the Author**

1. Is the manuscript technically sound, and do the data support the conclusions?

Reviewer #1: Yes

Reviewer #2: Yes

2. Has the statistical analysis been performed appropriately and rigorously? 

Reviewer #1: Yes

Reviewer #2: Yes

3. Have the authors made all data underlying the findings in their manuscript fully available?

Reviewer #1: Yes

Reviewer #2: No

4. Is the manuscript presented in an intelligible fashion and written in standard English?

Reviewer #1: Yes

Reviewer #2: No

5. Review Comments to the Author

Reviewer #1: The article entitled “Transcriptomic analysis of potato (Solanum tuberosum l.) tuber development reveals new insights into starch biosynthesis” presents the interesting study on potato tuber development and starch biosynthesis. The article is generally well-written and organized, presenting the methodology, results, and implications of the study in a clear and concise manner. The article will make a valuable contribution to the scientific literature in this field. However, I recommend to address the following comments and resubmit.

Comment 1: Line 13-15 can be rewritten as: "Many genes play major roles in different pathways, including carbohydrate metabolism during the potato tuber's life cycle," to increase the readability of the sentence.

Comment 2: In line 97, mention the pot mixture and whether it will influence tuber formation.

Comment 3: In line 98, mention the year in which the experiment was carried out. Also, mention how the light and dark conditions were maintained.

Comment 4: In line 100, mention the days after planting at which samples are taken for the study. Also, mention the size of the tubers.

Comment 5: Mention the agronomic performance of the cultivar and why a particular cultivar was preferred for the study.

Comment 6: In line 107, provide the company name/instrument name instead of the country.

Comment 7: Mention whether the statistical enrichment of DEGs was tested. If tested, mention the software used.

Comment 8: Mention whether the starch content of the potato tuber was analyzed at different stages.

Comment 9: In line 172-174, rearrange the sentence to read: "635 genes, followed by 315 genes, and then 234 genes."

Comment 10: All the protocols and software used should be mentioned only in the Materials and Methods section, not in the Results and Discussion.

Comment 11: Mention a few important DEGs identified in the study and their role in tuber formation.

Comment 12: Figures were given in the Microsoft PowerPoint presentation. It disrupts the quality of the figures. Hence, please provide them in JPEG or PNG format according to the journal's requirements.

Comment 13: Figure 1 is illegible. Please consider improving the quality for better readability.

Comment 14: Even though the manuscript is grammatically sound, the complexity of a few sentences may puzzle readers. Hence, consider using simpler sentences to attract more readers.

Reviewer #2: Introduction

The introduction prepared well and the recent literatures were cited

Lno; 55-57 include proper citations

Methodology

The methodology lacks many technical information and need to enhance the English

Mention year and station of the experiment conducted

Lno 105; rna quality checked in gel or nanodrop, mention properly

Lno 109; how 18 samples selected and what are all the six samples , is there any replication means mention properly

Lno 114; is it Trimmomatic version 3-6 or 3.6 ?

Lno123: what are all the different databases like NCBI nr, uniprot, etc

Concise the RT PCR portions

Results

Lno 156: method and software should not repeat here.

Give raw reads and filtered reads and the statistics table about the rnaseq reads

The obtained results not clearly explained and mainly not able to understand the concept

Need to improve the manuscript results section with proper English editing

Discussions

Well discussed with previous reports

Figures

Include significance in Fig 6

can try with the better visualization graphs like volcano plot, kegg and gene ontology bar graphs it helps to better understanding for readers.

Introduction:

The introduction is well-prepared and appropriately cites recent literature.

However, there is a need to provide proper citations for lines 55-57.

Methodology:

The methodology lacks technical details and requires improvement in English. Additionally, it would be helpful to mention the year and station where the experiment was conducted.

Line 105: Clarify whether RNA quality was checked using gel electrophoresis or a Nanodrop, and provide proper details.

Line 109: Explain how the 18 samples were selected and provide information about the six samples. Mention if there were any replications and provide proper clarification.

Line 114: Specify whether it is Trimmomatic version 3-6 or 3.6.

Line 123: List the different databases used

Concise the RT-PCR sections for better clarity.

Results:

Line 156: Ensure that the method and software are not repeated in this section. Provide information on raw reads and filtered reads, and include a statistics table for RNA-seq reads.

The obtained results are not clearly explained, and the underlying concept is hard to understand. Consider improving the English and structure of the results section.

Discussions:

The discussion is well-structured and references previous reports effectively.

Figures:

In Figure 6, include a description of the significance.

Consider using better visualization methods, such as volcano plots, KEGG and gene ontology bar graphs, to enhance reader understanding.

Overall:

Mainly need English editorial activity for this manuscript otherwise objective of the study and implementation are satisfactory.

6. PLOS authors have the option to publish the peer review history of their article (what does this mean?). If published, this will include your full peer review and any attached files.

Reviewer #1: **Yes: **Vijayakumar Eswaramoorthy

Reviewer #2: No

---

## [Author Response · Author response to Decision Letter 0]

19 Nov 2023

Reviewer #1: The article entitled “Transcriptomic analysis of potato (Solanum tuberosum.) tuber development reveals new insights into starch biosynthesis” presents the interesting study on potato tuber development and starch biosynthesis. The article is generally well-written and organized, presenting the methodology, results, and implications of the study in a clear and concise manner. The article will make a valuable contribution to the scientific literature in this field. However, I recommend to address the following comments and resubmit.

Authors’s Response: We are grateful to the respected reviewer for his/her gentle compliment as well as constructive comments. 

Comment 1: Line 13-15 can be rewritten as: "Many genes play major roles in different pathways, including carbohydrate metabolism during the potato tuber's life cycle," to increase the readability of the sentence.

Authors’s Response: Revised accordingly.

Comment 2: In line 97, mention the pot mixture and whether it will influence tuber formation.

Authors’s Response: Characteristics of the Marfona cultivar and soil type used for planting have been added to the manuscript. 

Comment 3: In line 98, mention the year in which the experiment was carried out. Also, mention how the light and dark conditions were maintained.

Authors’s Response: Experiment was conducted in 2016. Maintenance conditions have already been mentioned in Line 102-103.

Comment 4: In line 100, mention the days after planting at which samples are taken for the study. Also, mention the size of the tubers.

Authors’s Response: Sampling was done 6 weeks after planting. The size and the shape of samples were according to the description in the material and methods (figure 1).

Comment 5: Mention the agronomic performance of the cultivar and why a particular cultivar was preferred for the study.

Authors’s Response: Marfona cultivar is a popular early baker, high-yielding, resistance to dry rot, and susceptible to potato cyst nematode that is cultivated worldwide. Characteristics of the Marfona cultivar have been added to the text. 

Comment 6: In line 107, provide the company name/instrument name instead of the country.

Authors’s Response: Novogene Bioinformatics Technology is a company based in Beijing, China

Comment 7: Mention whether the statistical enrichment of DEGs was tested. If tested, mention the software used.

Authors’s Response: Thank you for the comment. DEGs are the output of Deseq2 package. As mentioned in the manuscript, The DESeq2 package calculate differential expression based on the negative binomial distribution model which is a strong statistical package based on R-language. By default, Deseq2 uses the Wald test (a chi-square test). 

Comment 8: Mention whether the starch content of the potato tuber was analyzed at different stages.

Authors’s Response: Appreciate for the comment. Unfortunately, we did not analyzed the starch content of developing tubers due to small size of tuber stages. We needed a delicate/sensitive kit in order to measure the small amount of starch present in tiny tubers. Unfortunately measuring carbohydrates with available manual protocols did not give us an accurate estimate of tiny tubers starch content. 

Comment 9: In line 172-174, rearrange the sentence to read: "635 genes, followed by 315 genes, and then 234 genes."

Authors’s Response: We appreciate. Revised accordingly.

Comment 10: All the protocols and software used should be mentioned only in the Materials and Methods section, not in the Results and Discussion.

Authors’s Response: Appreciate, redundant methods and softwares were deleted from the results and discussion sections.

Comment 11: Mention a few important DEGs identified in the study and their role in tuber formation.

Authors’s Response: We have already mentioned and discussed DEGs associated with starch biosynthesis in results and discussion sections.

Comment 12: Figures were given in the Microsoft PowerPoint presentation. It disrupts the quality of the figures. Hence, please provide them in JPEG or PNG format according to the journal's requirements.

Authors’s Response: Appreciate, we will try whether it can be possible to submit figure individually. 

Comment 13: Figure 1 is illegible. Please consider improving the quality for better readability.

Authors’s Response: We tried to enhance the quality of photos taken from tuber developmental stages. We think the quality is now acceptable. I wish we had taken the pictures with professional cameras, with a great regret we did not have one!!

Comment 14: Even though the manuscript is grammatically sound, the complexity of a few sentences may puzzle readers. Hence, consider using simpler sentences to attract more readers.

Authors’s Response: We are grateful to the respected reviewer. We had asked a native speaker to read the manuscript. We went through the manuscript and fixed some of the typo/grammar mistakes.

Reviewer #2: Introduction

The introduction prepared well and the recent literatures were cited

Lno; 55-57 include proper citations

Authors’s Response: Appreciate, appropriate citations were added.

Methodology

The methodology lacks many technical information and need to enhance the English

Mention year and station of the experiment conducted

Authors’s Response: Appreciate, Revised accordingly.

Lno 105; rna quality checked in gel or nanodrop, mention properly

Authors’s Response: Ambiguity was fixed. The new sentence meaning that the quality and quantity of RNA was double checked by both methods. 

Lno 109; how 18 samples selected and what are all the six samples, is there any replication means mention properly

Authors’s Response: RNA samples were selected for sequencing based on RIN number (RIN>7). Details have been incorporated in the manuscript.

Lno 114; is it Trimmomatic version 3-6 or 3.6 ?

Authors’s Response: Appreciate, we used Trimmomatic version 0.36 . 

Lno123: what are all the different databases like NCBI nr, uniprot, etc

Concise the RT PCR portions

Authors’s Response: We used Go and NCBI non-redundant databases. RT-qPCR details have been already mentioned in Line 153-155.

Results

Lno 156: method and software should not repeat here.

Authors’s Response: Appreciate, redundant methods and softwares were deleted from the results section.

Give raw reads and filtered reads and the statistics table about the rnaseq reads The obtained results not clearly explained and mainly not able to understand the concept Need to improve the manuscript results section with proper English editing

Authors’s Response:We are most grateful to the respected reviewer. We have now added table 1, showing the summary of read numbers based on the RNA-Seq data analysis from developing potato tuber stages.

Discussions

Well discussed with previous reports

Authors’s Response: Our best greetings for respected reviewer nice compliment. 

Figures

Include significance in Fig 6 can try with the better visualization graphs like volcano plot, kegg and gene ontology bar graphs it helps to better understanding for readers.

Authors’s Response: Expressions were compared with student t-test relative to SI. The figure 6 graph has now been modified with statistics signs showing the differences. We agree with the respected reviewer to use volcano plot, however, since the number of genes investigated are small, volcano plot does not look informative.

---

## [Decision Letter · Decision Letter 1]

18 Dec 2023

PONE-D-23-23589R1Transcriptomic analysis of potato (Solanum tuberosum L.) tuber development reveals new insights into starch biosynthesisPLOS ONE

Dear Dr. Nazarian‐Firouzabadi,

Thank you for submitting your manuscript to PLOS ONE. After careful consideration, we feel that it has merit but does not fully meet PLOS ONE’s publication criteria as it currently stands. Therefore, we invite you to submit a revised version of the manuscript that addresses the points raised during the review process.

**ACADEMIC EDITOR: **I suggest following the reviewer's suggestions and modifying the manuscript. Also, I recommend improving the english language. 

We look forward to receiving your revised manuscript.

Kind regards,

Karthikeyan Adhimoolam

Academic Editor

PLOS ONE

Journal Requirements:

Reviewers' comments:

Reviewer's Responses to Questions

**Comments to the Author**

1. If the authors have adequately addressed your comments raised in a previous round of review and you feel that this manuscript is now acceptable for publication, you may indicate that here to bypass the “Comments to the Author” section, enter your conflict of interest statement in the “Confidential to Editor” section, and submit your "Accept" recommendation.

Reviewer #1: All comments have been addressed

Reviewer #2: All comments have been addressed

2. Is the manuscript technically sound, and do the data support the conclusions?

Reviewer #1: Yes

Reviewer #2: Yes

3. Has the statistical analysis been performed appropriately and rigorously? 

Reviewer #1: Yes

Reviewer #2: Yes

4. Have the authors made all data underlying the findings in their manuscript fully available?

Reviewer #1: Yes

Reviewer #2: Yes

5. Is the manuscript presented in an intelligible fashion and written in standard English?

Reviewer #1: No

Reviewer #2: Yes

6. Review Comments to the Author

Reviewer #1: Dear Author,

I hope this message finds you well .I appreciate the valuable contribution your research. While the content of your article is robust and insightful, I have noted a few areas where attention to English language usage and clarity could enhance the overall quality of the manuscript. Please consider removing the words like "generally speaking".

I observed that there seems to be an overlap in the presentation of materials and methods, specifically pertaining to software, in both the Materials and Methods section and the Results section. To ensure clarity and adherence to the conventional structure of scientific manuscripts, it is recommended to confine details regarding materials, including software, strictly to the Materials and Methods section.

Kindly consider revisiting the Results section and refraining from introducing new materials and methods information. If any software or specific methods are crucial to understanding the results, it would be best to reference the earlier Materials and Methods section where they are appropriately detailed.

Reviewer #2: (No Response)

7. PLOS authors have the option to publish the peer review history of their article (what does this mean?). If published, this will include your full peer review and any attached files.

Reviewer #1: No

Reviewer #2: **Yes: **ARIHARASUTHARSAN G

---

## [Author Response · Author response to Decision Letter 1]

1 Jan 2024

ACADEMIC EDITOR: I suggest following the reviewer's suggestions and modifying the manuscript. Also, I recommend improving the english language.

Author’s Response: Thank you so much for your suggestion. We had our manuscript revised by a native speakers.

Reviewer #1: 

I hope this message finds you well .I appreciate the valuable contribution your research. While the content of your article is robust and insightful, I have noted a few areas where attention to English language usage and clarity could enhance the overall quality of the manuscript. Please consider removing the words like "generally speaking".

Author’s Response: we appreciate the respected reviewer comments. We had our manuscript revised by a native speakers.

I observed that there seems to be an overlap in the presentation of materials and methods, specifically pertaining to software, in both the Materials and Methods section and the Results section. To ensure clarity and adherence to the conventional structure of scientific manuscripts, it is recommended to confine details regarding materials, including software, strictly to the Materials and Methods section.

Author’s Response: Thank you so much for your constructing comment. We omitted the overlapping material and methods details from the results section.

Kindly consider revisiting the Results section and refraining from introducing new materials and methods information. If any software or specific methods are crucial to understanding the results, it would be best to reference the earlier Materials and Methods section where they are appropriately detailed.

Author’s Response: We went through the result section and deleted the unnecessary material and methods details.

---

## [Decision Letter · Decision Letter 2]

3 Jan 2024

Transcriptomic analysis of potato (Solanum tuberosum L.) tuber development reveals new insights into starch biosynthesis

PONE-D-23-23589R2

Dear Dr. Farhad,

We’re pleased to inform you that your manuscript has been judged scientifically suitable for publication and will be formally accepted for publication once it meets all outstanding technical requirements.

Kind regards,

Karthikeyan Adhimoolam

Academic Editor

PLOS ONE

Additional Editor Comments (optional):

Reviewers' comments:

Reviewer's Responses to Questions

**Comments to the Author**

1. If the authors have adequately addressed your comments raised in a previous round of review and you feel that this manuscript is now acceptable for publication, you may indicate that here to bypass the “Comments to the Author” section, enter your conflict of interest statement in the “Confidential to Editor” section, and submit your "Accept" recommendation.

Reviewer #1: All comments have been addressed

2. Is the manuscript technically sound, and do the data support the conclusions?

Reviewer #1: Yes

3. Has the statistical analysis been performed appropriately and rigorously? 

Reviewer #1: Yes

4. Have the authors made all data underlying the findings in their manuscript fully available?

Reviewer #1: Yes

5. Is the manuscript presented in an intelligible fashion and written in standard English?

Reviewer #1: Yes

6. Review Comments to the Author

Reviewer #1: (No Response)

7. PLOS authors have the option to publish the peer review history of their article (what does this mean?). If published, this will include your full peer review and any attached files.

Reviewer #1: **Yes: **Vijayakumar Eswaramoorthy

---

## [Editor Report · Acceptance letter]

27 Mar 2024

PONE-D-23-23589R2 

PLOS ONE

Dear Dr. Nazarian‐Firouzabadi, 

I'm pleased to inform you that your manuscript has been deemed suitable for publication in PLOS ONE. Congratulations! Your manuscript is now being handed over to our production team.

Kind regards, 

on behalf of

Dr. Karthikeyan Adhimoolam 

Academic Editor

PLOS ONE